# Lysine-14 acetylation of histone H3 in chromatin confers resistance to the deacetylase and demethylase activities of an epigenetic silencing complex

Mingxuan Wu[1,2,3], Dawn Hayward[3], Jay H Kalin[1,2,3], Yun Song[4], John WR Schwabe[4], Philip A Cole[1,2,3]*

[1]Division of Genetics, Department of Medicine, Brigham and Women's Hospital, Boston, United States; [2]Department of Biological Chemistry and Molecular Pharmacology, Harvard Medical School, Boston, United States; [3]Department of Pharmacology and Molecular Sciences, Johns Hopkins University School of Medicine, Baltimore, United States; [4]Leicester Institute of Structural and Chemical Biology, Department of Molecular and Cell Biology, University of Leicester, Leicester, United Kingdom

**Abstract** The core CoREST complex (LHC) contains histone deacetylase HDAC1 and histone demethylase LSD1 held together by the scaffold protein CoREST. Here, we analyze the purified LHC with modified peptide and reconstituted semisynthetic mononucleosome substrates. LHC demethylase activity toward methyl-Lys4 in histone H3 is strongly inhibited by H3 Lys14 acetylation, and this appears to be an intrinsic property of the LSD1 subunit. Moreover, the deacetylase selectivity of LHC unexpectedly shows a marked preference for H3 acetyl-Lys9 versus acetyl-Lys14 in nucleosome substrates but this selectivity is lost with isolated acetyl-Lys H3 protein. This diminished activity of LHC to Lys-14 deacetylation in nucleosomes is not merely due to steric accessibility based on the pattern of sensitivity of the LHC enzymatic complex to hydroxamic acid-mediated inhibition. Overall, these studies have revealed how a single Lys modification can confer a composite of resistance in chromatin to a key epigenetic enzyme complex involved in gene silencing.
DOI: https://doi.org/10.7554/eLife.37231.001

*For correspondence:
pacole@bwh.harvard.edu

## Introduction

Chromatin remodeling and histone modifications contribute to epigenetic regulation of gene expression in physiologic and disease processes. Enzymes that reversibly modify Lys residues in histones and other proteins are emerging therapeutic targets in cancer and other diseases. Such enzymes include the 'writers', Lys acetyltransferases (KATs) and Lys methyltransferases (KMTs) and 'erasers' histone deacetylases (HDACs) and lysine demethylases (KDMs) (*Cole, 2008*; *Helin and Dhanak, 2013*; *Shortt et al., 2017*). There are numerous members of each of these writer and eraser enzyme families and several of them have been studied intensively. Here, we investigate the catalytic activities of HDAC1 and LSD1.

HDAC1 is a Class I HDAC Zn hydrolase and has a broad range of targeted acetyl-lysine (Kac) sites in histones and non-histone proteins (*Cole, 2008*; *Falkenberg and Johnstone, 2014*; *Taunton et al., 1996*). HDAC1 and its closest paralog HDAC2 (sometimes the pair are called HDAC1/2) are typically found in multi-protein repressor complexes in the cell including the CoREST complex, the NuRD complex, the MiDAC complex and the Sin3a complex (*Bantscheff et al., 2011*;

*Itoh et al., 2015*; *Laugesen and Helin, 2014*). The CoREST complex which is investigated here is special because it contains HDAC1/2, LSD1, and the CoREST scaffold protein as core subunits in a 1:1:1 stoichiometry (*Kalin et al., 2018*). LSD1 is a flavin-dependent amine oxidase that demethylates mono- or di-methylated Lys4 in histone H3 (H3K4me1/2) (*Shi et al., 2004*). Because histone Kac and methyl H3K4 modifications are typically found in regions of transcriptionally active chromatin, the enzymatic activities of HDAC1 and LSD1 in the CoREST complex tend to synergistically result in gene silencing (*Kalin et al., 2018*; *Ooi and Wood, 2007*) (*Figure 1*). Recent efforts to develop small molecule inhibitors of HDAC1 and LSD1 individually and selectively targeted to the CoREST complex (*Arrowsmith et al., 2012*; *Høenfeldt et al., 2013*; *Kalin et al., 2018*; *Mohammad et al., 2015*; *Prusevich et al., 2014*) appear to show promise as anti-cancer agents.

Much of our current understanding of the enzymatic properties of HDAC1 and LSD1 has come from studying these enzymes as purified proteins or in multi-protein complexes besides that of the purified core CoREST complex LSD1/HDAC1/CoREST1 (LHC) (*Hayward and Cole, 2016*; *López et al., 2016*; *Marabelli et al., 2016*; *Millard et al., 2013*; *2017*; *Wagner et al., 2016*; *Watson et al., 2016*). Until recently, it has been challenging to make purified, active LHC although an effective method for this has recently been described (*Kalin et al., 2018*). In addition, enzymatic studies on LSD1 and HDAC1 have commonly been based on peptide substrates (*Hayward and Cole, 2016*; *López et al., 2016*; *Marabelli et al., 2016*; *Millard et al., 2013*; *2017*; *Wagner et al., 2016*; *Watson et al., 2016*) rather than the physiologic nucleosome substrates (*Dhall et al., 2017*; *Kim et al., 2015*) which are still technically challenging to prepare in site-specifically modified forms. One relatively new method to make semisynthetic histone H3 in N-tail modified forms involves mutant sortase, a mutant transpeptidase enzyme that can produce traceless H3 with site-specific post-translational modifications (*Piotukh et al., 2011*; *Ringel et al., 2015*). However, the

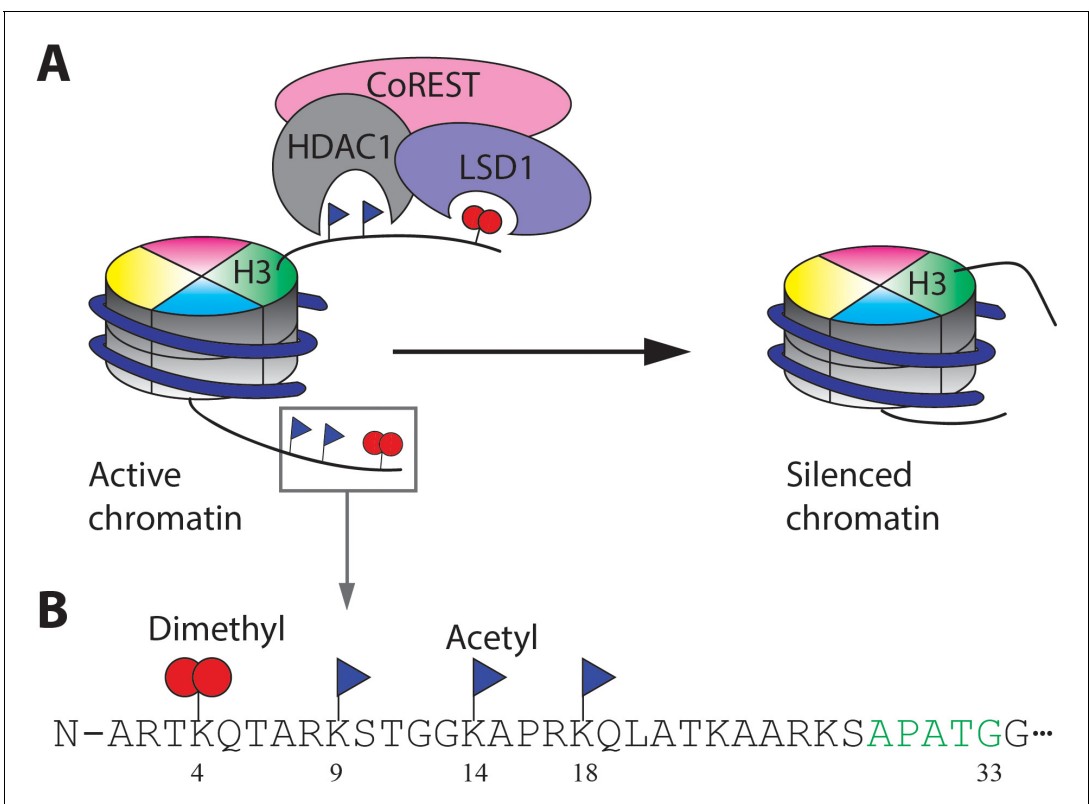

**Figure 1.** The CoREST complex and its H3 substrate histone modifications. (**A**) The CoREST complex (LHC) represses gene transcription by erasing histone modifications on nucleosomes. LSD1 demethylates H3K4me2 and HDAC1 deacetylates multiple Kac sites on histones which are activating marks. (**B**) The sequence of the N-terminal histone H3 tail and the Kme2 and Kac modifications studied in this paper. The short recognition sequence of F40 sortase is labelled in green.

DOI: https://doi.org/10.7554/eLife.37231.002

semisynthetic H3 ligation conversions are less than 50% which can complicate isolation and purification (*Piotukh et al., 2011*; *Ringel et al., 2015*).

Here, we use purified LHC along with various peptide and modified nucleosomal substrates to examine its deacetylase and demethylase properties. We introduce a technical improvement to the engineered sortase technique for generating semisynthetic histone H3s that allows for higher conversions. Using these tools, we have gleaned new insights into the substrate selectivity of the LHC's recognition and processing of chromatin substrates that enhance our understanding of histone code molecular recognition by a key multi-protein repressor complex.

## Results

### Characterization of the LHC demethylase activity with peptide substrates

We have employed the recently described method to obtain the purified core CoREST ternary protein complex (LHC) which contains HDAC1, LSD1, and CoREST1 as a heterotrimer in 1:1:1 stoichiometry (*Figure 2A*) (*Kalin et al., 2018*). This complex is stable for up to 2 weeks when stored at 4°C at a concentration of about 5 μM and freeze-thaws were avoided. LHC demethylase activity was determined with histone H3 peptide (aa1-21) substrate possessing K4me2 using a previously described coupled assay that measures hydrogen peroxide product formation (*Hayward and Cole, 2016*; *Marabelli et al., 2016*). Interestingly, using the unacetylated H3K4me2 tail peptide, we observed a bi-phasic demethylase activity with LHC that was not observed with purified GST-LSD1 as the catalyst (*Figure 2B*). The first phase is typically about 3–5 min in duration and after transition to the second phase, product formation is quite linear for at least 15 min (*Figure 2B*). Adjusting the concentration of the coupling enzyme or peroxide detection reagents did not eliminate the bi-phasic behavior (data not shown), suggesting it is an intrinsic property of the LHC-catalyzed reaction under these buffer conditions. By varying the peptide substrate concentration, we determined that the initial phase of the LHC-catalyzed reaction had lower apparent $K_m$ and $k_{cat}$ values compared to the second linear phase and that the $K_m$ value of the second phase was similar to that of GST-LSD1 (*Figure 2C* and *Figure 2—figure supplement 1*). Varying ionic strength, glycerol content, or inositol hexaphosphate (InsP$_6$), an HDAC complex stabilizer (*Watson et al., 2016*), was performed to see if the kinetic profile could be simplified by adjusting the reaction conditions. InsP$_6$ somewhat prolonged the initial phase (*Figure 2B*) but altering the buffer conditions did not eliminate the bi-phasic nature of the reaction. We considered the possibility that the LHC complex might be falling apart over the course of our assays, but size exclusion chromatography of the reaction mixture after 20 min showed that the complex was still intact (*Figure 2—figure supplement 2*).

It has been reported that LSD1/CoREST can bind to DNA in a DNA sequence independent fashion and that mononucleosomes containing extended DNA sequences are more efficiently demethylated by the LSD1/CoREST heterodimer (*Kim et al., 2015*; *Pilotto et al., 2015*; *Yang et al., 2006*). Interestingly, we found that the addition of 146 bp of dsDNA (the 601 Widom sequence [*Lowary and Widom, 1998*]) sharply accelerated LHC-mediated demethylation of the H3K4me2 tail peptide in a concentration- dependent way and also abolished the lag phase seen without DNA. (*Figure 2D*). The dose-response curve showed an EC$_{50}$ of 63 ± 15 ng/μL and a maximum activation of demethylation rate of 36-fold (*Figure 2—figure supplement 3*). We also tested a 4.9 kb circular DNA plasmid and a 60 bp linear dsDNA fragment. The plasmid DNA exhibited a similar level of stimulation of LHC catalyzed demethylation but the 60 bp dsDNA fragment was inert in these assay conditions (*Figure 2—figure supplement 3*). These results suggest that a large piece of DNA appears necessary for allosteric activation of LHC-mediated demethylation of H3K4me2 tail peptide, perhaps in part through a templating role.

BHC80 is another protein component that is well known to be associated with the CoREST complex (*Lan et al., 2007*; *Shi et al., 2005*). We used a readily available recombinant protein fragment of BHC80 and explored its effects on LSD1 demethylation. Interestingly, when a stoichiometric level of BHC80 (or higher) relative to the LHC concentration is added, the bi-phasic demethylase activity simplified with the second phase being markedly inhibited (*Figure 2E*). The BHC80 inhibitory effect appeared to saturate at 200 nM and it had only a slight impact on the initial phase of the demethylation reaction (*Figure 2E*, *Figure 2—figure supplement 3*).

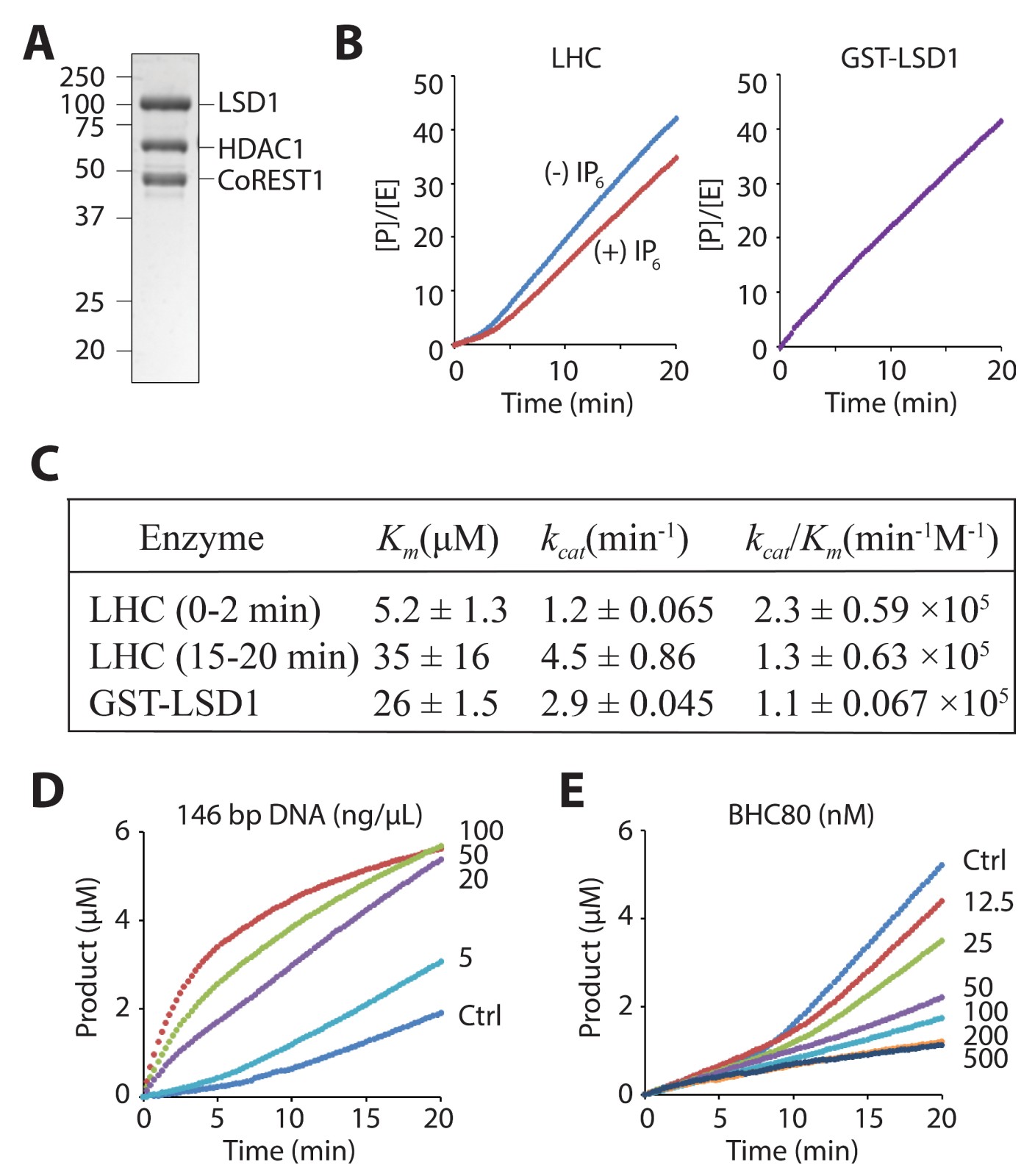

**Figure 2.** LHC purity and its demethylase activity with H3 tail peptide substrate. (**A**) Coomassie-stained SDS-PAGE of the purified ternary CoREST complex LHC. (**B**) Demethylation of 100 μM H3K4me2 H3 tail peptide (aa1-21) peptide by LHC (130 nM) exhibits a bi-phasic activity. In the presence of 100 μM InsP$_6$, the first phase becomes longer. In contrast, the demethylation by GST-LSD1 (88 nM) exhibits a linear activity that best matches the second phase of LHC as shown in panel C. The y-axis is shown as [Product]/[Enzyme]. (**C**) Summary of the steady-state kinetic parameters of peptide

*Figure 2 continued on next page*

*Figure 2 continued*

demethylation by LHC and GST-LSD1. (p<0.05 for $K_m$ and p<0.005 for $k_{cat}$ of the first phase vs. the second phase); (n = 3 for all measurements); kinetic parameters shown are ± S.E.M. (**D**) DNA accelerates the peptide demethylation by LHC (50 nM) in a dose dependent manner. (**E**) BHC80 modulates the peptide demethylation by LHC (100 nM), altering the bi-phasic activity (>100 nM).

DOI: https://doi.org/10.7554/eLife.37231.003

The following figure supplements are available for figure 2:

**Figure supplement 1.** Non-linear curve fitting to the Michaelis–Menten equation was used to calculate the observed $K_m$ and $k_{cat}$ values of LHC and GST-LSD1.

DOI: https://doi.org/10.7554/eLife.37231.004

**Figure supplement 2.** Size exclusion chromatography analysis of LHC after the demethylase reaction.

DOI: https://doi.org/10.7554/eLife.37231.005

**Figure supplement 3.** LHC demethylase activity with H3 tail peptide substrate in the presence of DNA or BHC80.

DOI: https://doi.org/10.7554/eLife.37231.006

The influences of $InsP_6$, DNA, and BHC80 on LHC are consistent with the idea that the bi-phase demethylase kinetics results from a conformational change of the LHC complex during catalysis, but further structural studies will be needed to clarify such a change. Regardless, we decided to carry out the subsequent LHC peptide demethylase assays in the presence of $InsP_6$ which can stabilize HDAC complexes (*Millard et al., 2013*; *2017*; *Watson et al., 2016*) and focused on the first linear 'lag' phase to characterize the demethylation kinetics with peptide substrates containing additional Lys acetylation modifications at specific sites.

LHC demethylation of histone H3K4me2 peptides that contained additional K9ac, K14ac or K18ac showed only minor effects on the demethylation rate, the largest of which was a 60% reduction with H3K4me2K14ac substrate (*Figure 3A*). Since the HDAC1 in LHC was in principle capable of deacetylating these substrates contemporaneously with demethylation, we examined the demethylation in the presence of 10 μM SAHA (Vorinostat), a broad spectrum HDAC inhibitor (*Grant et al., 2007*; *Richon et al., 1998*). The demethylase rates of the K9ac and K18ac containing peptide substrates were moderately reduced while the rate of the K14ac substrate was more sharply inhibited (*Figure 3A*) with 100 μM peptide substrate. These results indicate that deacetylation most likely precedes demethylation, consistent with the more rapid rate of deacetylase activity reported previously (*Kalin et al., 2018*). Steady-state kinetic analysis in the presence of SAHA revealed that the $K_m$ and $k_{cat}$ of LHC demethylase action on the H3K4me2K14ac substrate were dramatically decreased (~20 fold) relative to the non-acetylated substrate. In comparison, H3K4me2K9ac showed an intermediate

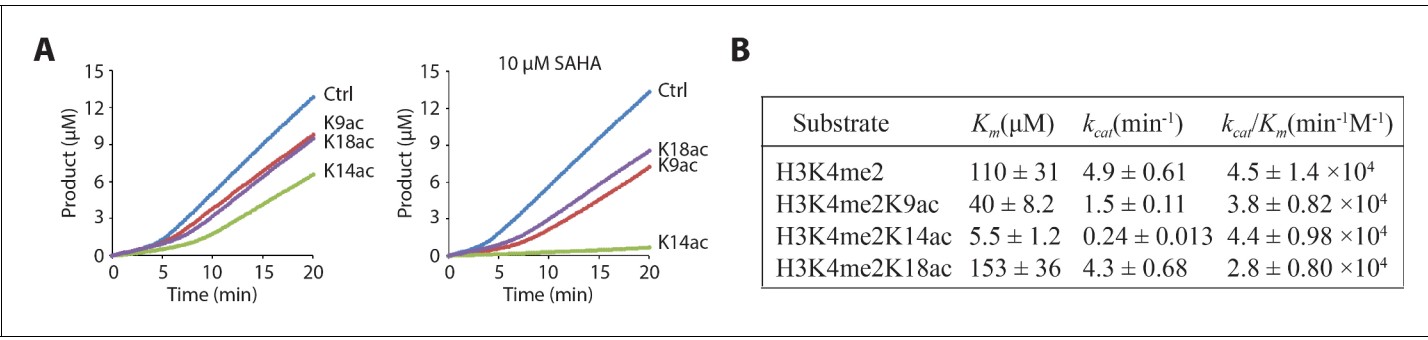

**Figure 3.** LHC demethylase activity with H3 tail peptide substrates containing Kac. (**A**) Comparison of LHC-catalyzed demethylation of 100 μM peptide substrate with H3K4me2 (Ctrl), H3K4me2K9ac (K9ac), H3K4me2K14ac (K14ac), and H3K4me2K18ac (K18ac) by 100 nM LHC. The four substrates were also analyzed in the presence of 10 μM SAHA. (**B**) Summary of the steady-state kinetic parameters based on the initial phase (0–2 min) of the reactions in the presence of SAHA. (p<0.005 for $K_m$ and $k_{cat}$ of H3K4me2 peptide vs. H3K4me2K14ac peptide substrates); (n = 3 for all measurements) error bars represent S.E.M.; kinetic parameters are shown ± S.E.M.

DOI: https://doi.org/10.7554/eLife.37231.007

The following figure supplement is available for figure 3:

**Figure supplement 1.** Isolated LSD1 demethylase activity with H3 tail peptide substrates containing Kac.

DOI: https://doi.org/10.7554/eLife.37231.008

effect with ~3 fold reductions in $k_{cat}$ and $K_m$ and H3K4me2K18ac was closely matched with the non-acetylated substrate in these LHC demethylase reactions (*Figure 3B* and *Figure 3—figure supplement 1*). Interestingly, the $k_{cat}/K_m$ values were closely matched among the various substrates suggesting that the reduced catalytic turnover appears to be associated with a stabilized enzyme-substrate ground state complex for H3K4me2K14ac. Prior biochemical studies with purified LSD1 have shown that substrate H3 tail acetylation is disruptive to catalysis (*Forneris et al., 2007*; *2006*). We have specifically investigated Lys9, Lys14, and Lys18 with GST-LSD1 here and found that Lys14 acetylation sharply inhibits H3K4me2 demethylation (*Figure 3—figure supplement 1*). These results suggest that LHC preserves an intrinsic property of LSD1 with respect to the H3K14ac inhibitory effects.

## Generating nucleosomes containing semi-synthetic site-specifically modified histone H3s

To further understand the enzymatic properties of LHC, we turned to site-specifically modified nucleosome substrates. To obtain the desired modified nucleosomes, we adapted F40 sortase, an engineered transpeptidase (*Piotukh et al., 2011*; *Ringel et al., 2015*), in combination with depsipeptide H3 tail peptide substrates to generate full length semisynthetic histone H3, F40 sortase has been used previously to generate semisynthetic H3 but the method suffers from poor ligation yield that is presumably related to the transpeptidation step which is thermodynamically balanced between starting material and product. Ester peptide substrates have the potential to overcome this limitation (*Williamson et al., 2012*). In our hands, the use of an ester linkage in the synthetic N-tail significantly improved the conversion of H3 from less than 40 to ~90% (*Figure 4A and B*). The more effective ligation facilitated purification of the full-length H3 away from the unligated H3 (*Figure 4—figure supplement 1*, overall isolated yield ~40% of purified semisynthetic H3) and the structures of the traceless semisynthetic histone H3s were confirmed by mass spectrometry (*Figure 4C*) and western blots (*Figure 4—figure supplement 1*). The semisynthetic H3s were then assembled along with the other core recombinant *Xenopus* histones H2A, H2B, and H4 to furnish histone octamers which in turn were used in mononucleosome reconstitution with 146 bp 601 DNA (*Figure 4A* and *Figure 4—figure supplement 1*).

## Characterization of the enzymatic activity of the CoREST complex using nucleosome substrates

We investigated the ability of LHC to demethylate H3K4me2 modified nucleosomes by monitoring H3K4me2 disappearance using western blot. However, we were unable to detect nucleosome demethylation even at the highest concentration of LHC employed (400 nM) with 100 nM nucleosome substrate over 3 hr. Given prior studies that DNA extension in nucleosomes can stimulate LSD1 activity with the LSD1/CoREST heterodimer (*Kim et al., 2015*), we generated H3K4me2 mononucleosomes containing 185 bp 601 DNA with 20 bp extensions on either end (20 + 145 + 20). With this nucleosome with longer DNA, we were able to observe LHC catalyzed demethylation, albeit at a slow rate (*Figure 5A* and *Figure 5—figure supplement 1*). To confirm the enzymatic dependence of this demethylation, we showed that the dual LSD1/HDAC1 inhibitor corin (*Kalin et al., 2018*) could block this demethylation activity. (*Figure 5B* and *Figure 5—figure supplement 1*). We estimate that the rate of this reaction is about 55-fold reduced relative to the corresponding tail peptide substrate. This low rate makes it difficult to perform in depth, quantitatively reliable enzymatic characterization.

We explored LHC's deacetylase activity with individually modified H3K9ac, H3K14ac, and H3K18ac 146 bp mononucleosome substrates (which also contained H3K4me2), monitoring deacetylase activity with site-specific anti-Kac Abs. These experiments revealed efficient deacetylation of 100 nM H3K9ac nucleosomes with 20 nM LHC within 20 min with comparatively slower rates measured with H3K18ac followed by H3K14ac (*Figure 6A and 7*). Selectivity was preserved with the corresponding triacetylated nucleosomes with K9ac > K18ac > K14 ac (*Figure 6B* and *Figure 6—figure supplement 2*). There was a ~7 fold rate differential between H3K9ac and K14ac. Interestingly, when purified H3 proteins were investigated as LHC substrates, they showed essentially identical rates for hydrolysis at the different positions (*Figure 6C* and *Figure 6—figure supplement 2*). As observed

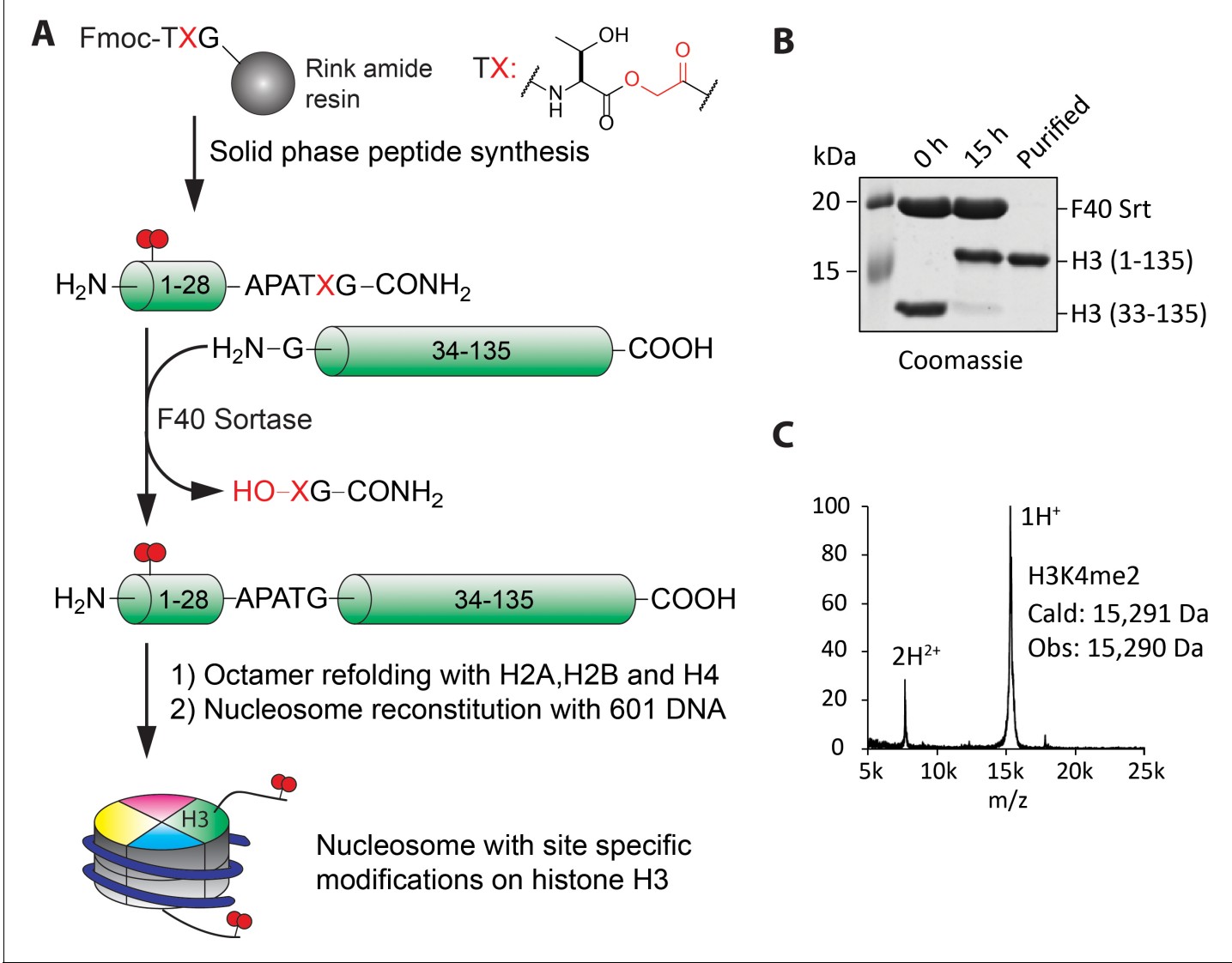

**Figure 4.** Preparation of nucleosomes with site-specific modifications using F40 sortase and depsipeptide H3 tails. (A) Scheme for the semisynthesis of histone H3 forms and their incorporation into nucleosomes. The ester bond between Thr32 and X33 enhances ligation conversion. (B) Coomassie-stained SDS-PAGE of the sortase mediated ligation and the purified histone H3 product containing H3K4me2. (C) MALDI mass spectrum of the semisynthetic H3K4me2 modified histone H3.

DOI: https://doi.org/10.7554/eLife.37231.009

The following figure supplements are available for figure 4:

**Figure supplement 1.** Purification and characterization of semisynthetic histone H3 and the reconstituted nucleosomes
DOI: https://doi.org/10.7554/eLife.37231.010
**Figure supplement 2.** MALDI mass spectra of the histone H3 peptides in this study.
DOI: https://doi.org/10.7554/eLife.37231.011
**Figure supplement 3.** MALDI mass spectra of the semi-synthetic histone H3s in this study.
DOI: https://doi.org/10.7554/eLife.37231.012
**Figure supplement 4.** $^1$H NMR of Fmoc-Thr(OtBu)-glycolic acid and Alloc-Asu(Hd-OtBu)-OH.
DOI: https://doi.org/10.7554/eLife.37231.013

with LHC catalyzed demethylation, the LHC deacetylation rates were much higher for the isolated H3 protein substrates compared with the nucleosome substrates (**Table 1**).

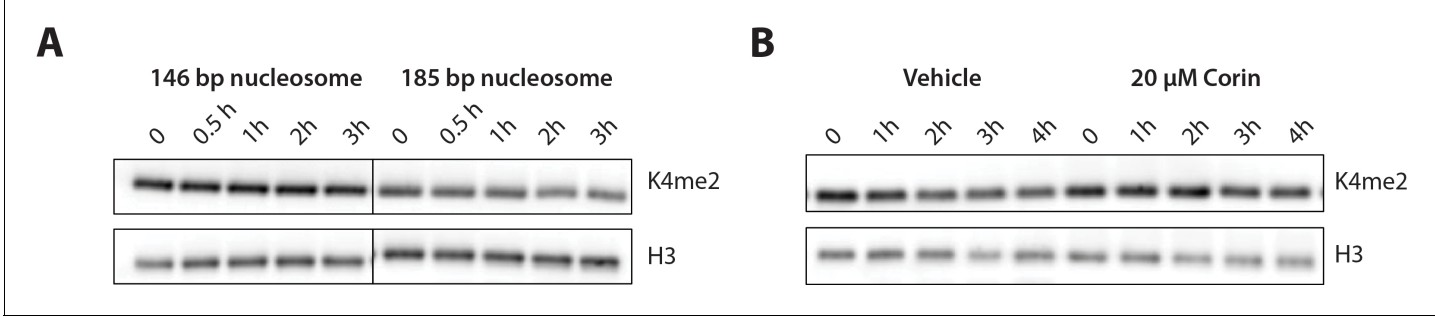

**Figure 5.** Demethylation of H3K4me2-modified nucleosomes by LHC. (**A**) Demethylation of 100 nM H3K4me2 modified nucleosomes with either 146 or 185 bp nucleosomal 601 DNA by 400 nM LHC. (**B**) Corin inhibits the demethylation of the 185 bp H3K4me2 modified nucleosome substrate. These assays were performed three times (n = 3) in A and twice (n = 2) in B on separate occasions.
DOI: https://doi.org/10.7554/eLife.37231.014

The following figure supplement is available for figure 5:

**Figure supplement 1.** Demethylation kinetics of H3K4me2 nucleosomes (100 nM) by LHC (400 nM) quantified by western blots.
DOI: https://doi.org/10.7554/eLife.37231.015

## Analysis of hydroxamic acid-Lys analogs with LHC

Hydroxamic acid (Hd) warheads have been employed in small molecule HDAC inhibitors such as SAHA (*Richon et al., 1998*) and also in the context of peptides as analogs of Lys residues (*Dose et al., 2016*). We envisaged that these Hd warheads if installed at the K9 and K14 positions of histone H3 could be useful to probe the basis of more facile deacetylation of nucleosomes containing K9ac versus K14ac. We hypothesized that H3K9Hd versus H3K14Hd might more potently inhibit LHC deacetylase activity if the deacetyaltion differences resulted from greater exposure of the nine position for LHC engagement. Therefore, we incorporated the Lys analog AsuHd (2-aminosuberic acid ω-hydroxamate) into semisynthetic histone H3 at the 9 and 14 positions (*Figure 7—figure supplement 1*). These Hd containing H3s were used in the assembly of 146 bp mononucleosomes. These H3K9Hd and H3K14Hd containing nucleosomes were probed as potential inhibitors of LHC catalyzed deacetylation of a fluorescent peptide substrate (*Figure 7A*). We found that both H3K9Hd and H3K14Hd containing nucleosomes could inhibit LHC deacetylase activity with sub-micromolar potencies (*Figure 7B*) whereas there was little effect of unmodified or acetylated mononuclesomes under these reaction conditions (*Figure 7C*). A detailed analysis showed that, as expected, H3K14Hd is a competitive inhibitor versus peptide substrate for the LHC reaction (*Figure 7D*). The $K_i$ values for H3K9Hd and H3K14Hd containing nucleosomes were 60 nM and 40 nM, respectively for LHC inhibition (*Figure 7F*). The corresponding H3 Hd-containing proteins and peptides were about 30-fold lower (*Figure 7E* and *Figure 7—figure supplement 1*). The similar potencies of these Hd nucleosomes suggest similar accessibility of these nucleosomes for the LHC HDAC1 active site. Thus, we deduce that the preference of LHC for deacetylating K9ac versus K14ac in nucleosomes is not a simple difference in exposure but must derive from more subtle interactions such as precise orientation of the acetyl-Lys in the active site.

## Discussion

We have carried out a comparative analysis of the core CoREST complex, LHC, and its enzymatic processing of site-specially modified H3 peptide/protein as well as nucleosome substrates. In general, the demethylase and deacetylase activities of LHC are far greater when processing the simpler H3 peptide or protein substrates relative to the nucleosomes, consistent with the idea that the H3 tails are not as sterically available to the LHC active sites in nucleosomes. As the LHC serves a corepressor role in gene regulation, we speculate that the recruitment of LHC by a DNA-binding transcription factor such as REST (*Burg et al., 2015*; *Qureshi et al., 2010*; *Zhou et al., 2013*) would significantly enhance LHC targeting to specific regions of chromatin. In this way, the CoREST complex would show precise regional control in silencing gene expression through its enzymatic activities.

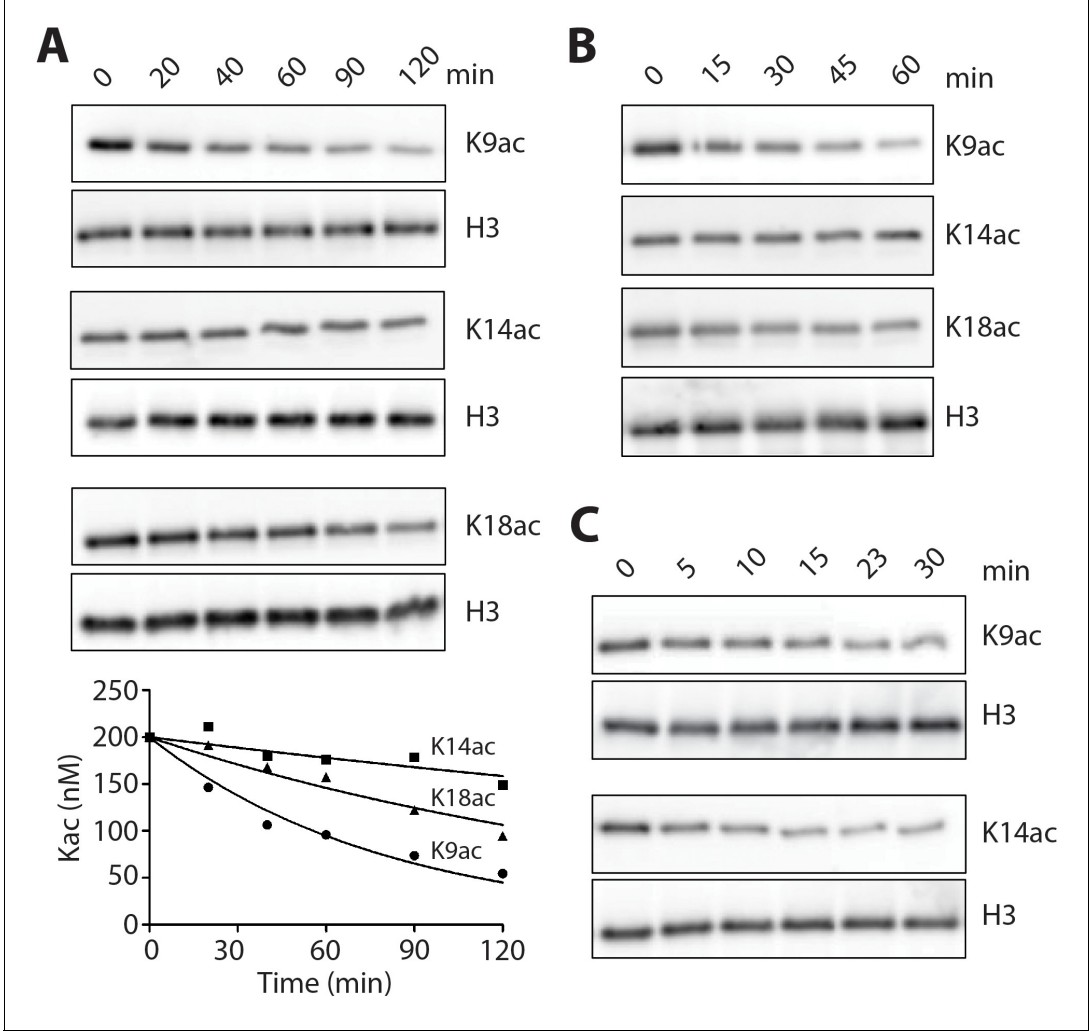

**Figure 6.** LHC deacetylation of nucleosomes and isolated histone H3 containing Kac. (**A**) The deacetylation of 100 nM monoacetylated nucleosomes by 20 nM LHC. The quantified bands of Kac are measured in the plot below. (**B**) Deacetylation of 200 nM triacetylated H3K9acK14acK18ac nucleosomes by 40 nM LHC. (**C**) The deacetylation of 1 µM monoacetylated semisynthetic histone H3s possessing either K9ac or K14ac by 1 nM LHC. These assays were performed at three times (n = 3) in A and B and twice (n = 2) in C on separate occasions.

DOI: https://doi.org/10.7554/eLife.37231.016

The following figure supplements are available for figure 6:

**Figure supplement 1.** LHC and isolated HDAC1 deacetylation of nucleosomes and isolated histone H3 containing Kac.

DOI: https://doi.org/10.7554/eLife.37231.017

**Figure supplement 2.** Quantification and curve fitting of the deacetylation assays.

DOI: https://doi.org/10.7554/eLife.37231.018

The demethylase activity of LHC retains the selectivity of free LSD1 and its reduced activity toward H3 tails that are marked concomitantly with acetylation, particularly at Lys14. We think that the K14Ac inhibitory effect is likely through an intramolecular mechanism. We believe this because the peptide substrate length requirement for efficient processing by isolated LSD1 is a minimum of 20 aa of the H3 tail starting from Ala1 (*Culhane and Cole, 2007*). An X-ray crystal structure of LSD1 in complex with an H3 peptide substrate analog shows electron density for the tail peptide extending beyond Lys14 in the complex with LSD1 (*Forneris et al., 2007*). This structure shows that the vicinity of LSD1 near the Lys14 sidechain is negatively charged, and this may account for the reduced activity of the Lys14 acetylation substrate.

The revelation here that Lys14 deacetylation by LHC in nucleosomes is sluggish compared with deacetylation from Lys9 and Lys18 was unanticipated. In fact, we show here that LHC's deacetylase

**Table 1.** Summary of the deacetylation turnover (V/[E]) from data in *Figure 6* and *Figure 6—figure supplement 1* (p<0.001 for V/[E] of K9ac nucleosome vs. K14ac nucleosome substrates in both the monoacetylated and triacetylated assays); (n = 3 for all measurements except n = 2 for LHC on H3 and HDAC1 on nucleosomes; kinetic values shown are ± S.E.M.

| | Substrate | V/[E] (min$^{-1}$) | | |
| | | K9ac | K14ac | K18ac |
|---|---|---|---|---|
| LHC | Kac Nucleosome | 0.10 ± 0.0084 | 0.014 ± 0.0073 | 0.048 ± 0.011 |
| | Kac H3 | 31 ± 3.9 | 35 ± 8.5 | - |
| | 3Kac Nucleosome | 0.11 ± 0.0071 | 0.019 ± 0.0033 | 0.058 ± 0.0022 |
| | 3Kac H3 | 31 ± 4.3 | 35 ± 2.7 | - |
| HDAC1 | Kac Nucleosome | <0.005 | <0.005 | - |
| | Kac H3 | 10 ± 0.89 | 12 ± 0.26 | - |

DOI: https://doi.org/10.7554/eLife.37231.019

selectivity with nucleosomes is not intrinsic for acetylated H3 tail peptides or even full length acetylated H3 protein. The potential biological implications of this chromatin selectivity are noteworthy since they suggest that the CoREST complex's gene silencing function will be limited against chromatin marked by H3K14ac but more impactful in chromatin lacking this mark. As LHC will be slow to demethylate H3K4me2 when an adjacent H3K14ac is present, and H3K14ac is particularly resistant

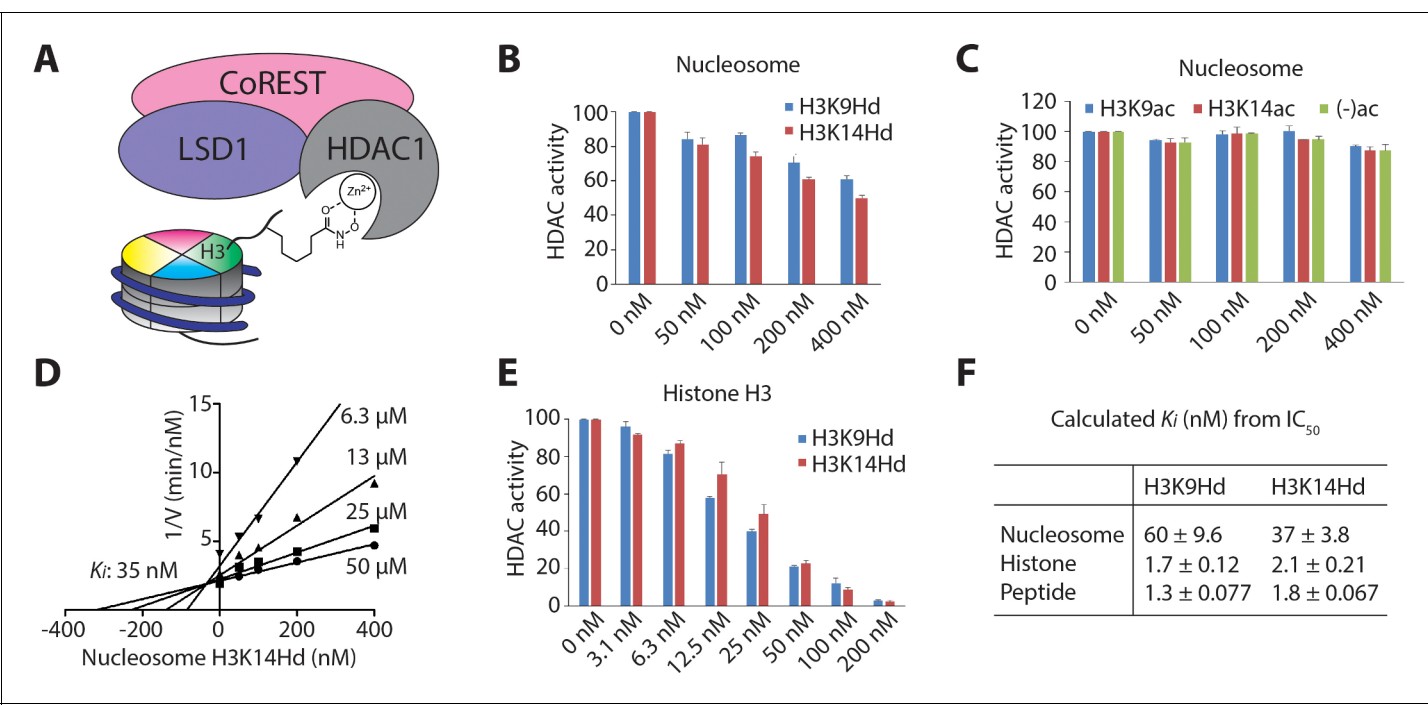

**Figure 7.** Nucleosomes with hydroxamic acid Lys analogs were potent competitive inhibitors of LHC-catalyzed deacetylation of Kac peptide substrate. (A) A model for nucleosomes containing hydroxamic acid acetyl-Lys analogs (KHd) on the H3 tail and coordinating to the active site $Zn^{2+}$ of HDAC1 in LHC. (B) Dose-dependent inhibition of LHC catalyzed peptide deacetylation by KHd-modified nucleosomes. (C) Kac and unacetylated nucleosomes (up to 400 nM) do not inhibit LHC catalyzed peptide deacetylation. (D) Dixon plot of LHC inhibition by H3K14Hd-modified nucleosomes shows competitive inhibition versus peptide deacetylation with $K_i$ = 35 nM. (E) Dose-dependent inhibition of LHC catalyzed peptide deacetylation by KHd-modified semisynthetic histone H3s. (F) Summary of the $K_i$ values calculated from IC$_{50}$ values, (n = 3 for all measurements); values shown are ± S.E.M.

DOI: https://doi.org/10.7554/eLife.37231.020

The following figure supplement is available for figure 7:

**Figure supplement 1.** Quantification and curve fitting of the inhibition of the deacetylation assays.

DOI: https://doi.org/10.7554/eLife.37231.021

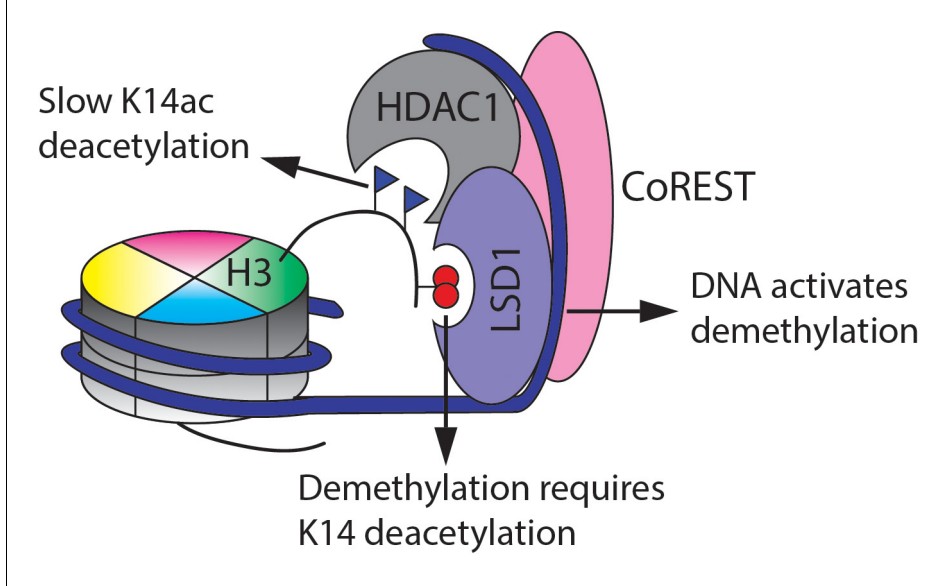

**Figure 8.** Summary of the findings here of the enzymatic activities of the CoREST complex with a modified nucleosome substrate.
DOI: https://doi.org/10.7554/eLife.37231.022

to LHC catalyzed deacetylation, the CoREST complex is hypothesized to be a poorly effective corespressor to switch off genes with an abundance of H3K4me2/H3K14ac in promoter or enhancer regions.

Whether other corepressor complexes containing HDAC1 such as Sin3a or NuRD are more efficient at deacetylating H3K14ac in chromatin is uncertain. However, it is notable that global H3K14ac, relative to H3K9ac, is minimally changed after cells are treated with the HDAC1-3 inhibitor entinostat (*Kalin et al., 2018*; *Schölz et al., 2015*), suggesting that the nucleosomal deacetylase selectivity of LHC may be a general property of class I HDAC complexes. It has been proposed that H3K14ac can modulate the structure of the nucleosomal H3 tail as suggested by a computational study (*Ikebe et al., 2016*) which could render such nucleosomes resistant to LHC deacetylase action. Evidence against this possibility is that the triacetylated H3 tail nucleosomes studied here showed similar deacetylase rates compared with the mono-acetylated nucleosomes.

We applied the hydroxamic acid (Hd) substitutions as potential transition state analogs that would correlate with the catalytic efficiencies of deacetylation at the H3K9ac and H3K14ac positions. This correlation in affinity and catalytic efficiency was not observed experimentally. We have considered a few reasons for this: (1) the hydroxamic acid interaction with the Zn does not capture the positioning of the acetamide of the substrate (that is, Hd is not really binding as a transition state analog); (2) the Hd analog sidechain is somewhat longer than that of acetyl-Lys so that the histone H3 tail backbone is further removed from the HDAC substrate binding surface; (3) the conformation of the H3 tail is different with the hydroxamic acids versus the acetyl-Lys sidechains and thus makes different contacts with the HDAC substrate binding surface. At this stage, we do not have information that can distinguish among these possibilities. In future experiments, we plan to substitute specific residues proximal to K9 and K14 in the context of nucleosomes to see if there are specific tail residues that govern the LHC deacetylation selectivities.

These studies also clearly show that deacetylation, which is much more rapidly catalyzed than demethylation, is likely to be the first step in dynamic gene silencing by LHC. This was perhaps best exemplified by the difference in histone H3 peptide demethylation being only modestly affected by K14ac unless SAHA was present to block the deacetylase activity. The bi-phasic nature of the LHC demethylase activity remains incompletely understood. We believe it suggests that there are at least two different conformations of the LHC complex with altered rate-determining steps associated with the two states, but more detailed structural studies will be needed to clarify this possibility. How

DNA and BHC influence these potential conformations will also be interesting to pursue in future work. We hypothesize that DNA may be an allosteric activator of LSD1 and may also play a role as a template to bring substrate and enzyme closer together. The hydroxamic acid nucleosomes introduced in this study to probe H3 tail accessibility may also prove useful in enhancing structural characterization of a nucleosome-LHC complex by increasing the stability of this association (*Pilotto et al., 2015*).

From a technical standpoint, we have combined the use of F40 sortase and depsipeptide H3 tails to improve the production of semisynthetic H3. We believe that for generating N-terminally modified H3s, this approach offers advantages over native chemical ligation (*Holt and Muir, 2015*; *Seenaiah et al., 2015*), nonsense suppression (*Neumann et al., 2008*; *Wang et al., 2017*), or other methods (*Chalker et al., 2012*) in combining high yield and efficiency of incoporating tails with multiple and diverse modifications without the need for Cys mutation/desulfurization.

## Conclusion

In this study, we applied chemical and biochemical tools to study how LSD1 and HDAC1 in the context of the CoREST complex act on nucleosomes to modify histone marks. We find that H3K4me2/H3K14ac is a privileged pair of modifications that is especially resistant to CoREST enzymatic processing. These findings provide a new framework for understanding how gene regulation may be controlled by specific histone patterns of post-translational modifications.

# Materials and methods

The source data for this manuscript have been deposited in Dryad (DOI: https://doi.org/10.5061/dryad.413tm83; *Wu et al., 2018*).

## Reagents

Amino acids were purchased from Novabiochem of EMD Millipore (Darmstadt, Germany) and Chem-Impex (Wood Dale, IL). Anti-histone H3 (Abcam Cat# ab1791, RRID:AB_302613), anti-H3K4me2 (Abcam Cat# ab32356, RRID:AB_732924) and anti-H3K9ac (Abcam Cat# ab32129, RRID:AB_732920) antibodies and BHC80 were purchased from Abcam (Cambridge, MA). Anti-H3K14ac (Millipore Cat# 07–353, RRID:AB_310545) and anti-H3K18ac (Millipore Cat# 07–354, RRID:AB_441945) antibodies were purchased from EMD Millipore (Burlington, MA). HRP-linked anti-rabbit antibody (Cell Signaling Technology Cat# 7074, RRID:AB_2099233) was purchased from Cell Signaling Technology (Danvers, MA). HEK293F cell line (RRID:CVCL_6642) was purchased from Thermo Fisher Scientific (Waltham, MA) and shown to be negative for mycoplasma. His-FLAG-HDAC1 and Fluorogenic HDAC1 assay kit was purchased from BPS Bioscience (San Diego, CA).

## Peptide synthesis

Histone H3 peptides (aa 1–21) were synthesized using the Fmoc strategy on a PS3 peptide synthesizer (Gyros protein technologies, Tucson, AZ). Syntheses started with 0.1 mmol Fmoc-Ala-Wang resin and Fmoc was deprotected by 5 mL 20% piperidine in dimethylformamide (DMF) for 10 min twice. Next, the resin was coupled twice with 0.4 mmol Fmoc-amino acid, 1.6 mmol N-methylmorpholine (NMM) and 0.375 mmol HATU in 4 mL DMF for 90 min at room temperature. After the coupling steps, the resin was washed with DMF prior to subsequent cycles. After all amino acid couplings were complete, the resin was transferred to a reaction column and deprotected with 20% piperidine and then washed with DMF and $CH_2Cl_2$. Peptide cleavage from resin and protective group removal was performed by treatment with reagent K (5% phenol, 5% $H_2O$, 5% thioanisole and 2.5% ethanedithiol in trifluoroacetic acid (TFA)) for 3 hr at room temperature. The crude peptide product was precipitated with cold diethyl ether and dried by nitrogen gas flow and then purified by reversed phase HPLC with a Varian Dynamax Microsorb 100–5 C18 column (250 × 21.4 mm, 5 μm). The gradient condition is 5% $CH_3CN$/0.05%TFA in $H_2O$/0.05%TFA for 2 min and a linear gradient from 5% $CH_3CN$/0.05%TFA to 32% $CH_3CN$/0.05%TFA over 25 min, flow rate: 10 mL/min.

Histone H3 depsipeptides (aa 1–34) were synthesized using the Fmoc strategy on a Prelude peptide synthesizer (Gyros protein technologies, Tucson, AZ) with the same coupling and deprotection conditions described for the PS3 reactions with standard Fmoc amino acids. Syntheses started with 0.1 mmol Fmoc-Rink Amide resin. After the first cycle of Fmoc-Gly-OH, 0.2 mmol depsipeptide

Fmoc-Thr(OtBu)-glycolic acid (prepared as described in a published method (*Williamson et al., 2012*), *Figure 4—figure supplement 4*) was coupled. The resin was then transferred to a Prelude reaction vessel and the remaining couplings were performed using automated synthesis except those involving specialty amino acids including Fmoc-Lys(me2)-OH and Fmoc-Lys(Ac)-OH. For specialty couplings, the resin was transferred to a reaction column and 0.2 mmol amino acid was used in the overnight coupling reactions to maximize efficiencies. After completion of all amino acid couplings and deprotection of Fmoc, the resin was transferred to a reaction column and washed with DMF and $CH_2Cl_2$. The resin was treated with 2.5% $H_2O$ and 2.5% TIPS in TFA for cleavage and deprotection over 3 hr at room temperature. The crude peptide product was precipitated with cold diethyl ether and dried by nitrogen gas flow and then purified by reversed phase HPLC using the same method. The fractions were combined and lyophilized to yield usually 55–75 mg (16%–21%) H3 depsipeptide based on the 0.1 mmol scale.

For synthesis of the peptides used for H3K4me2K9Hd and H3K4me2K14Hd semisynthesis, the coupling step for the hydroxamic acid containing amino acids was performed manually with 0.12 mmol Alloc-Asu(Hd-OtBu)-OH (prepared according to a previous method (*Dose et al., 2016*), *Figure 4—figure supplement 4*) with an overnight coupling reaction to maximize efficiencies. After washing with DMF, 0.4 mmol acetic anhydride and 1.6 mmol NMM were added to the mixture for 1 hr to cap residual free amine. After washes with DMF and $CH_2Cl_2$, the resin was treated with 0.4 mmol Pd(PPh$_3$)$_4$ in 8 mL CHCl$_3$/AcOH/NMM (40:2:1) for 3 hr and the resin was then washed with 0.5% sodium diethyldithiocarbonate trihydrate followed by 0.4 M NMM in DMF. The Alloc-deprotected resin was then transferred back to the Prelude for the remaining amino acid couplings. After completion of amino acid couplings and final Fmoc deprotection, the resin was transferred to a reaction column and washed with DMF and $CH_2Cl_2$ and then the peptide was deblocked and cleaved from the resin by treatment with 2.5% $H_2O$ and 2.5% TIPS in TFA for 16 hr at room temperature. The crude peptide was precipitated using cold diethyl ether and dried by nitrogen gas flow and then purified by reversed phase HPLC with the same method. The fractions were combined and lyophilized to yield usually 25–30 mg (7%–9%) H3 depsipeptide based on the 0.1 mmol scale.

Synthesized peptides (matrixed with $\alpha$-cyano-4-hydroxycinnamic acid) were characterized by Voyager DE-STR MALDI-TOF at the Mass Spectrometry and Proteomics Core of Johns Hopkins University School of Medicine or at the Molecular Biology Core Facilities of Dana Farber Cancer Institute (*Figure 4—figure supplement 2*).

H3K4me2(1-21): $[M + H]^+$ calculated for $C_{96}H_{177}N_{36}O_{28}^+$ 2282.4, found 2282.5.
H3K4me2K9ac(1-21): $[M + H]^+$ calculated for $C_{98}H_{179}N_{36}O_{29}^+$ 2324.4, found 2324.6.
H3K4me2K14ac(1-21): $[M + H]^+$ calculated for $C_{98}H_{179}N_{36}O_{29}^+$ 2324.4, found 2324.6.
H3K4me2K18ac(1-21): $[M + H]^+$ calculated for $C_{98}H_{179}N_{36}O_{29}^+$ 2324.4, found 2324.6.
H3K4me2(1-34): $[M + H]^+$ calculated for $C_{146}H_{265}N_{54}O_{44}^+$ 3479.0, found 3478.8.
H3K4me2K9ac(1-34): $[M + H]^+$ calculated for $C_{148}H_{267}N_{54}O_{45}^+$ 3521.0, found 3521.0.
H3K4me2K14ac(1-34): $[M + H]^+$ calculated for $C_{148}H_{267}N_{54}O_{45}^+$ 3521.0, found 3521.7.
H3K4me2K18ac(1-34): $[M + H]^+$ calculated for $C_{148}H_{267}N_{54}O_{45}^+$ 3521.0, found 3521.7.
H3K4me2K9acK14acK18ac(1-34): $[M + H]^+$ calculated for $C_{152}H_{271}N_{54}O_{47}^+$ 3605.0, found 3605.6.
H3K9ac(1-34): $[M + H]^+$ calculated for $C_{146}H_{263}N_{54}O_{45}^+$ 3493.0, found 3493.6.
H3K4me2K9Hd(1-34): $[M + H]^+$ calculated for $C_{148}H_{267}N_{54}O_{46}^+$ 3537.0, found 3537.8.
H3K4me2K14Hd(1-34): $[M + H]^+$ calculated for $C_{148}H_{267}N_{54}O_{46}^+$ 3537.0, found 3537.8.

## Expression and purification of recombinant F40 sortase

F40 sortase production was performed using a previously reported method with some modifications (*Piotukh et al., 2011*). The DNA plasmid pET21-F40-srtA was transformed into BL21 *E. coli* and after plating a single colony was cultivated in LB media with 100 mg/L ampicillin at 37°C. After reaching an OD600 of 0.6, sortase expression was induced by adding 0.25 mM isopropyl β-D-thiogalactoside (IPTG) and further cultivated for 4 hr at 30°C. The cells were harvested by centrifugation at 5,000 g for 20 min and the cell pellet was resuspended in 20 mM Tris pH 8.0, 0.1% Triton X-100 and Roche protease inhibitor cocktail. The suspended cells were passed twice through a french press cell disrupter (Thermo electron corporation) at 1,500 psi and then the lysate was centrifuged at 20,000 g for 20 min. The resulting supernatant was applied to a column with prewashed Ni Sepharose 6 Fast Flow (GE Healthcare) and the column was washed with 20 mM Tris-HCl, 500 mM NaCl at pH 7.5. The column was then treated with elution buffers containing increasing

concentrations of imidazole (up to 250 mM) in 20 mM Tris pH 7.5 and 150 mM NaCl. Fractions were analyzed using coomassie-stained SDS-PAGE and those containing F40 sortase were combined and dialyzed against 50 mM Tris-HCl, 150 mM NaCl and 5 mM CaCl$_2$, pH 7.5 (3.5K MWCO, Thermo) followed by concentration using an Amicon Ultra spin column (10K MWCO, EMD Millipore) to 50 mg/mL. The purified proteins were stored at −80°C until usage.

## Expression of core histones

The *X. laevis* histones H2A, H2B, H3 and H4 were expressed in *E. coli* as previously described (*Luger et al., 1999*). The expression and purification procedure for the globular H3 (gH3) fragment (aa_33–135) was modified from a previous report (*Piotukh et al., 2011*). The DNA plasmid pET23-gH3 construct was transformed into the BL21 strain of *E. coli* and after plating a single colony was cultivated in LB media with 100 mg/L ampicillin at 37°C. After reaching an OD600 of 0.6, histone gH3 expression was induced with 0.5 mM IPTG, and the cells were cultivated for another 3 hr at 37°C. The cells were harvested by centrifugation at 5,000 g for 20 min and the cell-pellet was resuspended in histone wash buffer (50 mM Tris-HCl pH 7.5, 100 mM NaCl, 1 mM EDTA, 5 mM 2-mercaptoethanol (BME) and 0.2 mM phenylmethylsulfonyl fluoride (PMSF)) with 1% Triton X-100. The cells were lysed by three passes through a french press cell disrupter at 1,500 psi. The lysate was centrifuged at 20,000 g for 20 min, the supernatants were discarded and the pellets were washed with histone wash buffer with 1% Triton X-100 once and histone wash buffer without Triton X-100 twice. The pellets were then treated with 7 M guanidinium hydrochloride, 20 mM Tris pH 7.5 and 10 mM DTT. After centrifugation at 20,000 g for 15 min, the supernatant was dialyzed three times against IEX (ion exchange) buffer (7 M urea, 10 mM Tris pH 7.8, 1 mM EDTA, 0.2 mM PMSF and 5 mM BME). The resulting gH3 solution in urea buffer was diluted 5-fold by IEX buffer and loaded on to a tandem Q-SP column (GE healthcare, HiTrap Q HP and HiTrap SP HP, 5 mL) equilibrated with IEX buffer containing 100 mM NaCl. After loading, the column was eluted with a gradient of IEX buffer containing increasing NaCl from 100 mM to 500 mM. The fractions containing gH3 were collected and dialyzed against 2 mM BME (3.5K MWCO, Spectra/Por) followed by concentration using an Amicon Ultra spin column (3K MWCO, EMD Millipore) to 100 µM. The purified gH3 solution was stored at −80°C until usage.

## Semisynthesis of histone H3 using F40 sortase

The histone H3 depsipeptide (aa1-34, 1.2 mM) and gH3 (aa33-135, 70 µM) were mixed in reaction buffer (50 mM HEPES pH 7.5, 150 mM NaCl and 5 mM CaCl$_2$) and then F40 sortase (300 µM) was added. After incubation at 37°C overnight, the crude mixture was dialyzed against 1 L IEX buffer (7 M urea, 10 mM Tris pH 7.8, 1 mM EDTA and 5 mM BME). The resulting H3 solution was loaded on to a SP column (GE healthcare, HiTrap SP HP, 1 mL) equilibrated with IEX buffer containing 100 mM NaCl. After loading, the column was eluted with a gradient of IEX buffer containing increasing NaCl from 100 to 500 mM (*Figure 4—figure supplement 1*). The fractions containing ligated H3 were collected and dialyzed against 2 mM BME (3.5K MWCO, Spectra/Por) followed by concentration using an Amicon Ultra spin column (3K MWCO, EMD Millipore). The purified H3 was lyophilized to a white powder and stored at −80°C until usage. A typical reaction volume is 3.2 mL with 2.6 mg gH3 and the usual yield of the ligated H3 is 1.5 mg (46%). Semi-synthesized histone H3s (matrixed with sinapinic Acid) were characterized by Voyager DE-STR MALDI-TOF at the Mass Spectrometry and Proteomics Core of Johns Hopkins University School of Medicine or at the Molecular Biology Core Facilities of Dana Farber Cancer Institute (*Figure 4—figure supplement 3*).

H3K4me2: $[M + H]^+$ calculated for $C_{672}H_{1136}N_{215}O_{186}S_3^+$ 15291, found 15290.
H3K4me2K9ac: $[M + H]^+$ calculated for $C_{674}H_{1138}N_{215}O_{187}S_3^+$ 15333, found 15331.
H3K4me2K14ac: $[M + H]^+$ calculated for $C_{674}H_{1138}N_{215}O_{187}S_3^+$ 15333, found 15329.
H3K4me2K18ac: $[M + H]^+$ calculated for $C_{674}H_{1138}N_{215}O_{187}S_3^+$ 15333, found 15333.
H3K4me2K9acK14acK18ac: $[M + H]^+$ calculated for $C_{678}H_{1142}N_{215}O_{189}S_3^+$ 15417, found 15416.
H3K9ac: $[M + H]^+$ calculated for $C_{672}H_{1134}N_{215}O_{187}S_3^+$ 15304, found 15305.
H3K4me2K9Hd: $[M + H]^+$ calculated for $C_{674}H_{1138}N_{215}O_{188}S_3^+$ 15349, found 15350.
H3K4me2K14Hd: $[M + H]^+$ calculated for $C_{674}H_{1138}N_{215}O_{188}S_3^+$ 15349, found 15346.

## Octamer refolding and nucleosome reconstitution

Octamer refolding and nucleosome assembly were performed as previously reported(*Luger et al., 1999*). Briefly, the core histone proteins H2A, H2B, H3 and H4 were dissolved in unfolding buffer (7 M guanidine, 20 mM Tris pH 7.5 and 10 mM DTT) and mixed at molar ratio 1.1:1.1:1:1. After dialysis against high salt buffer (20 mM Tris 7.5, 2.0 M NaCl, 1 mM EDTA and 5 mM BME), the octamer was purified by size exclusion chromatography. The DNA used for these nucleosomes, 146 and 185 bp Widom 601 DNA, was prepared by methods previously reported (*Luger et al., 1999*). Briefly, the 146 bp DNA was obtained from restriction digests of an established DNA plasmid cultured in and isolated from *E. coli* followed and purified by size fractionation precipitation with polyethylene glycol. The 185 bp DNA was amplified by PCR from the DNA template and purified by preparative polyacrylamide gel electrophoresis. The histone octamer and DNA were mixed at a 1:1 molar ratio at high salt buffer (10 mM Tris 7.5, 2.0 M KCl, 1 mM EDTA and 1 mM DTT), and the mixture was gradually dialyzed to low salt buffer (10 mM Tris 7.5, 0.25 M KCl, 1 mM EDTA and 1 mM DTT). The resulting mixture was subjected to HPLC with a Waters instrument with 1525 binary pump and a 2489 UV-Vis detector. The gradient condition with a TEKgel column is 0% TES600/100% TES 250 for 12 min and a linear gradient from 25% TES 600% to 75% TES 600 over 30 min, flow rate: 1 mL/min. The fractions were collected and dialyzed to a storage buffer (20 mM Tris 7.5 and 1 mM DTT) and concentrated to 5–10 μM using an Amicon Ultra spin column (10K MWCO, EMD Millipore). Due to the zinc ion in the CoREST complex, we did not use the traditional TCS buffer that includes 1 mM EDTA to store the nucleosomes.

## Expression and purification of the CoREST complex

We followed the expression methods and purification protocols previously reported by Portolano *et al* (*Portolano et al., 2014*) and Kalin *et al* (*Kalin et al., 2018*). Briefly, HEK293F cells were co-transfected with HDAC1, CoREST1 and LSD1 DNA plasmids, and the complex was purified from the cell lysate by immunoaffinity enrichment and gel filtration. The purified complex was concentrated to 3–5 μM and stored at 4°C for up to 3 weeks.

## Kinetic analysis of the demethylation of H3 peptide substrates

The peroxidase-coupled kinetics measurements of the demethylation of H3 peptide substrate by the CoREST complex was performed as previously described (*Prusevich et al., 2014*). Briefly, the H3K4me2 peptide is mixed with 0.1 mM 4-aminoantipyrine, 1 mM 3,5-dichloro-2-hydroxybenzenesulfonic acid, 0.04 mg/mL horseradish peroxidase in 50 mM HEPES pH 7.5 followed by the addition of the CoREST complex to initialize the reaction at 25°C. Absorbance changes were measured at 515 nm and the product formation was quantified using the extinction coefficient of 26,000 $M^{-1}$. The data in the initial linear phase and second linear phase were analyzed separately using linear regression and GraphPad Prism five software to determine reaction rates.

## Analysis of demethylation of H3K4me2 nucleosomes

H3K4me2 nucleosomes (100 nM, 146 bp or 185 bp) were treated with LHC (400 nM) in a buffer with 50 mM HEPES 7.5 and 0.2 mg/mL BSA at 25°C. At each time point, 10 μL aliquots were mixed with 2 μL 80 mM EDTA and 4 μL 4x SDS-PAGE gel loading buffer. The samples were boiled for 5 min at 95°C, and resolved by 15% SDS-PAGE. After transferring to nitrocellulose, H3K4me2 and total H3 were detected by western blot with specific antibodies on separate gels. Western blot bands were visualized by ECL and quantified using ImageJ software, and the data were fit to a single phase exponential decay curve using GraphPad Prism five software.

## Analysis of deacetylation of acetylated nucleosomes and acetylated histone H3s

The H3Kac nucleosomes (100 nM) or histone H3s (1.0 μM) were treated with LHC (1–20 nM) or His-FLAG-HDAC1 (5 nM) in reaction buffer containing 50 mM HEPES 7.5, 100 mM KCl, 100 μM InsP$_6$, and 0.2 mg/mL BSA at 37°C. At each time point, 10 μL aliquots were mixed with 2 μL 80 mM EDTA and 4 μL 4x SDS-PAGE gel loading buffer. The samples were boiled for 5 min at 95°C, and resolved by 15% SDS-PAGE. After transferring to nitrocellulose, H3K9ac, H3K14ac, H3K18ac and total H3 were detected by western blot with specific antibodies on separate gels. Western blot bands were

visualized by ECL and quantified using ImageJ software and the data was fit to a single phase exponential decay curve using GraphPad Prism five software. In general, these experiments were done with histone H3 or nucleosomes that also contained H3K4me2. As our earlier work suggested that demethylation of K4me2 is very slow compared with deacetylation, we hypothesized that the H3K4 modification would not perturb deacetylation by LHC. In fact, we prepared H3K9ac nucleosomes lacking H3K4me2 and these showed similar behavior to those that contained H3K4me2 (please compare *Figure 6—figure supplement 1* with *Figure 6A*).

## Fluorogenic HDAC1 deacetylation assay and inhibition

LHC (1 nM) was pretreated with hydroxamic acid (K9Hd or K14Hd) containing nucleosomes (50–400 nM), histone H3 (3–200 nM), or H3 peptide (3–800 nM) or control nucleosomes in reaction buffer (50 mM HEPES 7.5, 100 mM KCl, 100 µM InsP$_6$ and 0.2 mg/mL BSA) for 30 min in ice. Next, the HDAC fluorogenic peptide substrate (50 µM) was added and the reaction mixtures were incubated at 37°C for 15 min. The HDAC Developer (2x) was added to quench the deacetylation reactions and develop the fluorescence signal for 15 min at room temperature. The samples were analyzed in 96-well plates by a fluorescent plate reader with excitation at a wavelength of 365 nm and detection of emitted light in the range of 450 nm. The data was analyzed by non-linear curve fitting (log(inhibitor) vs response) using GraphPad Prism five software to determine the IC$_{50}$. Ki was calculated from the equation Ki=IC$_{50}$/([S]/Km +1) assuming a competitive inhibition model that was demonstrated by Dixon plot for K14Hd nucleosomes (*Figure 7D*).

## Acknowledgements

We would like to thank Dirk Schwarzer (Tübingen) for helpful ideas and sharing the F-40 sortase and the globular histone H3 DNA constructs, Cynthia Wolberger (Johns Hopkins) for the core histone DNA constructs, Song Tan (Penn State) for the 146 bp 601 DNA construct, Robert Levendosky and Greg Bowman (Johns Hopkins) for the 601 DNA template and assistance with preparing the 185 bp 601 DNA. We would also like to thank Yana Li and the eukaryotic tissue culture facility (Johns Hopkins) for assistance with LHC expression. JWRS is a Wellcome Trust Senior Investigator (grant WT100237) and Royal Society Wolfson Research Merit Award Holder. He is also supported by a Biotechnology and Biological Sciences Research Council Project Grant BB/J009598/1 as well as funds from 4SC. We acknowledge NIH (GM62437), the FAMRI Foundation, and the V Foundation for financial support.

## Additional information

### Competing interests

Philip A Cole: Senior editor, *eLife*. The other authors declare that no competing interests exist.

### Funding

| Funder | Grant reference number | Author |
| --- | --- | --- |
| National Institute of General Medical Sciences | GM62437 | Philip A Cole |
| Flight Attendant Medical Research Institute | Center of Excellence | Philip A Cole |
| V Foundation for Cancer Research | Program Grant | Philip A Cole |
| Wellcome Trust | WT100237 | John WR Schwabe |
| Wolfson Foundation | | John WR Schwabe |
| Biotechnology and Biological Sciences Research Council | | John WR Schwabe |

The funders had no role in study design, data collection and interpretation, or the decision to submit the work for publication.

## Author contributions
Mingxuan Wu, Data curation, Software, Formal analysis, Validation, Investigation, Methodology, Writing—original draft, Writing—review and editing; Dawn Hayward, Jay H Kalin, Data curation, Software, Formal analysis, Validation, Writing—review and editing; Yun Song, Resources, Methodology, Writing—review and editing; John WR Schwabe, Conceptualization, Resources, Supervision, Funding acquisition, Methodology, Project administration, Writing—review and editing; Philip A Cole, Conceptualization, Supervision, Funding acquisition, Writing—original draft, Project administration, Writing—review and editing

## Author ORCIDs
Mingxuan Wu (iD) http://orcid.org/0000-0003-4721-0825
Jay H Kalin (iD) http://orcid.org/0000-0002-8747-6022
Yun Song (iD) http://orcid.org/0000-0002-6966-3801
John WR Schwabe (iD) http://orcid.org/0000-0003-2865-4383
Philip A Cole (iD) http://orcid.org/0000-0001-6873-7824

## Decision letter and Author response
Decision letter https://doi.org/10.7554/eLife.37231.027
Author response https://doi.org/10.7554/eLife.37231.028

# Additional files

## Supplementary files
• Transparent reporting form
DOI: https://doi.org/10.7554/eLife.37231.023

## Data availability
All data generated or analyses during this study have been deposited in Dryad.

The following dataset was generated:

| Author(s) | Year | Dataset title | Dataset URL | Database, license, and accessibility information |
|---|---|---|---|---|
| Wu M, Dawn H, Jay H Kalin, Yun Song, John WR Schwabe, Philip A Cole | 2018 | Data from: Lysine-14 acetylation of histone H3 in chromatin confers resistance to the deacetylase and demethylase activities of an epigenetic silencing complex | http://dx.doi.org/10.5061/dryad.413tm83 | Available at Dryad Digital Repository under a CC0 Public Domain Dedication |

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
