## [Decision Letter]

Thank you for submitting your article "Lysine-14 Acetylation of Histone H3 in Chromatin Confers Resistance to the CoREST Complex" for consideration by *eLife*. Your article has been favorably reviewed by two peer reviewers, and the evaluation has been overseen by Reviewing Editor Wilfred van der Donk and John Kuriyan as the Senior Editor. The following individual involved in review of your submission has agreed to reveal his identity: Champak Chatterjee (Reviewer #1).

The reviewers have discussed the reviews with one another and the Reviewing Editor has drafted this decision to help you prepare a revised submission.

Summary:

In this study, the authors use semisynthetic methylated and acetylated nucleosomes as substrates to study the effect of H3 tail acetylation on H3K4me2 demethylation by the LSD1-CoREST-HDAC1 core complex (LHC). Using nucleosomes with well-defined tail modifications, the authors make several key observations that advance our understanding of LCH function. LSD1 demethylase activity on K4me2 was found to be strongly inhibited by H3 Lys14ac. Furthermore, the deacetylase selectivity of LHC shows a marked preference for H3K9ac. LHC displayed biphasic kinetics of demethylation, which is not seen with LSD1 alone, and a BHC80 protein fragment eliminates the second phase of LSD1 activity. In addition to reporting on the activity of LHC with defined substrates, the authors also report a sortase ligation strategy to generate site-specifically modified H3 protein in high yield that was used for preparation of semisynthetic mononucleosome. This technology is suggested to be complementary and sometimes superior to existing methods. Based on the observations a reasonable model is proposed for LCH function. This study furthers our understanding of histone modification in nucleosomes and provides a potentially powerful new method that may allow the community to more readily prepare these complex substrates.

Please revise the manuscript in response to the points made below. Note that no new experimental work is requested.

Important points to consider:

1) The authors mention that the sortagging strategy to provide site-specifically modified H3 is more facile than other existing methods. However, they do not report yields for the semisynthesis, nor the synthesis of the depsipeptide. This renders it hard to evaluate such a concrete conclusion.

2) The effects of DNA or BHC80 on LHC activity are interesting. Is the activation by DNA an allosteric effect? It seems that DNA facilitates the transition of a low-activity conformation of LHC to a high-activity conformation. On the other hand, BHC80 stabilizes LHC in the low-activity conformation state.

3) The presence of H3K14ac is shown to have an inhibitory effect on LSD1's K4me2 demethylase activity. Does this have to be an intramolecular mechanism, or is it possible that two substrates bind so that K14ac intermolecularly affects K4 demethylation? Does the deacetylated nucleosome dissociate from the LHC complex prior to the demethylation reaction?

4) Quantification of total H3 and H3 PTMs (methylation and acetylation) in each gel lane was undertaken with different sets of antibodies. It is unclear from the text how the blots were visualized as both antibodies presumably target the same H3 band and may occlude each other to varying degrees. Some more detail in the quantification of WB data relative to H3 would be useful.

5) Mononucleosome is a poor substrate for LHC compared to H3 peptide. Does the mononucleosome become a better substrate upon adding 146bp DNA to the reaction system, in a similar way as shown in Figure 2D?

6) The authors state that "there was little effect of unmodified or acetylated mononuclesomes", but there are no data on unmodified mononucleosomes in Figure 7C.

7) Based on the data in Figure 7F, it appears that HDAC1 equally recognizes K9ac and K14ac. However, the authors also find that LHC prefers deacetylating K9ac over K14ac in nucleosomes (Figure 6). Can the authors add more discussion explaining these results?

---

## [Author Response]

Important points to consider:1) The authors mention that the sortagging strategy to provide site-specifically modified H3 is more facile than other existing methods. However, they do not report yields for the semisynthesis, nor the synthesis of the depsipeptide. This renders it hard to evaluate such a concrete conclusion.

We regret that we did not report the typical yields. We have now added these to the Materials and methods section and note that the purified semisynthetic histone H3 yield is 46% based on the limiting substrate (gH3). The standard purified depsipeptide yields are 16-21% on a 0.1 mmol scale whereas those containing the hydroxamic acid analog are 7-9% performed on the same scale. We also hope that we did not come across as denigrating existing methods to generate histone H3 which have their own strengths.

2) The effects of DNA or BHC80 on LHC activity are interesting. Is the activation by DNA an allosteric effect? It seems that DNA facilitates the transition of a low-activity conformation of LHC to a high-activity conformation. On the other hand, BHC80 stabilizes LHC in the low-activity conformation state.

We also believe that the effects of DNA and BHC80 on LHC demethylase action are interesting, despite the fact that we don't understand their structural basis. It seems likely to us that the activation of the LHC demethylase activity by DNA involves an allosteric effect but it may also involve a templating role of some kind. We have added these speculations to the Discussion.

3) The presence of H3K14ac is shown to have an inhibitory effect on LSD1's K4me2 demethylase activity. Does this have to be an intramolecular mechanism, or is it possible that two substrates bind so that K14ac intermolecularly affects K4 demethylation? Does the deacetylated nucleosome dissociate from the LHC complex prior to the demethylation reaction?

We think that the K14Ac inhibitory effect is likely through an intramolecular mechanism. We believe this because the peptide substrate length requirement for efficient processing by isolated LSD1 is a minimum of 20 aa of the H3 tail starting from Ala1. An X-ray crystal structure of LSD1 in complex with an H3 peptide substrate analog shows electron density for the tail peptide extending beyond Lys14 in the complex with LSD1. This structure shows that the vicinity of LSD1 near the Lys14 sidechain is negatively charged, and this may account for LSD1's reduced activity toward the Lys14 acetylation substrate. We have added this speculation to the Discussion. Although we have not measured the dissociation rate of the deacetylated nucleosome from LHC, we think that it is likely to dissociate fairly rapidly relative to the demethylase reaction, given how slow the latter is.

4) Quantification of total H3 and H3 PTMs (methylation and acetylation) in each gel lane was undertaken with different sets of antibodies. It is unclear from the text how the blots were visualized as both antibodies presumably target the same H3 band and may occlude each other to varying degrees. Some more detail in the quantification of WB data relative to H3 would be useful.

The analysis of methylation and acetylation by western blot shown in Figure 5 and 6 did indeed involve separate analysis with specific antibodies for the particular methylation or acetylation sites as well as total H3. There was no stripping of blots and reprobing in these experiments. Instead, each reaction mixture was run on sufficient scale and divided into portions that were analyzed to assess total H3 and PTM-modified H3 on separate blots. We have added this information to the Materials and methods section. Since these reactions involved purified nucleosomes from the same stock, the issue of equal loading from lane to lane is less of a concern then, for example, performing such experiments after isolation of proteins from crude extracts. However, inclusion of the total H3 controls supports the idea that even loading was achieved and that proteolytic breakdown over the time scale of the experiment is not a problem under these conditions.

5) Mononucleosome is a poor substrate for LHC compared to H3 peptide. Does the mononucleosome become a better substrate upon adding 146bp DNA to the reaction system, in a similar way as shown in Figure 2D?

We did a pilot experiment along these lines and 146 bp DNA addition in trans had no effect on demethylase action of LHC on nucleosomes. However, we are reluctant to include this or pursue this experiment in depth because adding excess large DNA could theoretically cause instability of the nucleosomes rendering interpretation complex.

6) The authors state that "there was little effect of unmodified or acetylated mononuclesomes", but there are no data on unmodified mononucleosomes in Figure 7C.

Thanks for spotting this error. We have now added the data for the non-acetylated tail to this panel which was omitted inadvertently from the manuscript (it has also been included in the data deposition in Dryad).

7) Based on the data in Figure 7F, it appears that HDAC1 equally recognizes K9ac and K14ac. However, the authors also find that LHC prefers deacetylating K9ac over K14ac in nucleosomes (Figure 6). Can the authors add more discussion explaining these results?

We applied the hydroxamic acid (Hd) substitutions as potential transition state analogs that would correlate with the catalytic efficiencies of deacetylation at the H3K9ac and H3K14ac positions. This correlation in affinity and catalytic efficiency was not observed experimentally. We have considered a few reasons for this: 1) the hydroxamic acid interaction with the Zn does not capture the positioning of the acetamide of the substrate (that is, Hd is not really binding as a transition state analog); 2) the Hd analog sidechain is somewhat longer than that of acetyl-Lys so that the histone H3 tail backbone is further removed from the HDAC substrate binding surface; 3) the conformation of the H3 tail is different with the hydroxamic acids versus the acetyl-Lys sidechains and thus makes different contacts with the HDAC substrate binding surface. At this stage, we do not have information that can distinguish among these possibilities. In future experiments, we plan to substitute specific residues proximal to K9 and K14 in the context of nucleosomes to see if there are specific tail residues that govern the LHC deacetylation selectivities.